# Sentiment Analysis of Arabic Course Reviews of a Saudi University Using Support Vector Machine

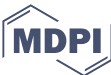

**Ali Louati [1,2,\*], Hassen Louati [3], Elham Kariri [1], Fahd Alaskar [1] and Abdulaziz Alotaibi [1]**

1   Information Systems, College of Computer Engineering and Sciences, Prince Sattam bin Abdulaziz University, Al-Kharj 11942, Saudi Arabia; e.kariri@psau.edu.sa (E.K.); 439051073@std.psau.edu.sa (F.A.); 439051779@std.psau.edu.sa (A.A.)
2   SMART Lab, ISG Tunis, University of Tunis, Le Bardo, Tunis 2000, Tunisia
3   College of Information Technology, Kingdom University, Riffa 40434, Bahrain; h.louati@ku.edu.bh
\*   Correspondence: a.louati@psau.edu.sa

**Abstract:** This study presents the development of a sentimental analysis system for high education students using Arabic text. There is a gap in the literature concerning understanding the perceptions and opinions of students in Saudi Arabia Universities regarding their education beyond COVID-19. The proposed SVM Sentimental Analysis for Arabic Students' Course Reviews (SVM-SAA-SCR) algorithm is a general framework that involves collecting student reviews, preprocessing them, and using a machine learning model to classify them as positive, negative, or neutral. The suggested technique for preprocessing and classifying reviews includes steps such as collecting data, removing irrelevant information, tokenizing, removing stop words, stemming or lemmatization, and using pre-trained sentiment analysis models. The classifier is trained using the SVM algorithm and performance is evaluated using metrics such as accuracy, precision, and recall. Fine-tuning is done by adjusting parameters such as kernel type and regularization strength to optimize performance. A real dataset provided by the deanship of quality at Prince Sattam bin Abdulaziz University (PSAU) is used and contains students' opinions on various aspects of their education. We also compared our algorithm with CAMeLBERT, a state-of-the-art Dialectal Arabic model. Our findings show that while the CAMeLBERT model classified 70.48% of the reviews as positive, our algorithm classified 69.62% as positive which proves the efficiency of the suggested SVM-SAA-SCR. The results of the proposed model provide valuable insights into the challenges and obstacles faced by Arab Universities post-COVID-19 and can help to improve their educational experience.

**Keywords:** sentimental analysis; support vector machine; Arabic course review; PSAU

## 1. Introduction

The significance of continuous assessment in educational institutions is paramount. It is instrumental for gauging student needs and perspectives, as well as for assessing the efficacy of educational programs. Accreditation agencies, acknowledging this, have increasingly emphasized outcome-based assessments to align educational programs with institutional goals and learning objectives. Additionally, the evolving job market and industry demands necessitate perpetual enhancement of educational programs to ensure students are well-prepared for professional challenges. Historically, one of the most informative methods to assess student learning experiences has been through the collection and analysis of feedback. Feedback serves as a mirror reflecting past behaviors and as a predictor for future improvements [1]. Particularly in education, feedback is critical for comprehending student engagement with curriculum and pedagogical strategies [2]. When students provide formative feedback, they offer essential insights into the course's strengths, the obstacles they encounter, and the effectiveness of teaching methods. Surveys have traditionally been a primary method for collecting student feedback. They can be

deployed anonymously, encouraging candidness, and can range from rating-based questions to open-ended textual responses. While rating-based surveys yield quantitative data, they often lack the depth of understanding provided by textual feedback [3]. Recognizing the limitations of conventional feedback analysis, we propose a novel approach using machine learning (ML) algorithms to analyze student feedback obtained from an online survey platform. Current research indicates several methods for sentiment analysis: rule-based, lexicon-based, and machine learning-based [4]. We adopt a machine learning-based approach, acknowledging its growing importance in extracting sentiment from textual data [5]. The Support Vector Machine (SVM) method is selected for its effectiveness in text classification and its robustness in handling high-dimensional data [6]. However, its application in analyzing educational feedback, particularly in the context of improving program effectiveness, is less explored. This paper aims to bridge this gap by detailing the use of SVM in the sentiment analysis of student feedback. The main contribution of this paper is the development of a methodological framework for extracting and analyzing sentiment from student feedback using SVM. This framework has the potential to enhance educational experiences by providing educators with nuanced insights into student sentiment, thereby improving the design and delivery of educational programs. Moreover, this study pioneers in developing a sentiment analysis system specifically for analyzing high-education students' feedback in Arabic text, addressing a significant gap in current literature. Our key contributions are the creation of the SVM-SAA-SC algorithm for processing Arabic course reviews; a comprehensive preprocessing and classification technique involving tokenization, stop word removal, and stemming; the employment of SVM for classification fine-tuned for optimal performance; the analysis of a substantive dataset from PSAU reflecting diverse educational aspects; and benchmarking against the advanced CAMeLBERT-DA SA Model to validate our approach. This endeavor is poised to uncover valuable insights into the unique challenges Arab students faced during the transition to online learning post-COVID-19 and to enhance the educational experience within Saudi Arabia's higher education system.

## 2. Related Work

### 2.1. Sentimental Analysis in Social Media

This section explores the Use of Sentiment Analysis in understanding social media Opinions and emotions. The use of sentiment analysis in understanding the opinions and emotions expressed on social media platforms, particularly Twitter, has been widely studied in recent years. Research in this area has focused on various topics, such as COVID-19 tweets [7], shared mobility services [8], and emotional intelligence in social media posts [9].

In [7], sentiment analysis was applied to COVID-19 tweets in English using TextBlob and TwitterScrapper in Python, achieving an accuracy of 79.34%. The study also applied sentiment analysis to medical documents in English using SVM and KNN, achieving an accuracy of 80%. However, the scope of this study was limited to textual data from Twitter and only a limited number of languages were considered.

Similarly, in [8], sentiment analysis was applied to tweets related to shared mobility services using a sentiment-emotion detection model, survey, and manual annotation and labeling. The study found that 67% of respondents preferred passenger reservation services over other available routes such as public transportation. The majority of tweets were neutral in the pre-COVID-19 era, but there was a 2% change in trend from positive to negative in the COVID-19 era. The study also found that words and frequency distribution of phrases gathered from significant Twitter data were common phrases such as "Uber", "Rideshare", "Drivers", and "Lyft".

In [9], sentiment analysis was applied to social media posts using preprocessing, feature extraction, matrix formulation, and classification. The study found that LRA-DNN achieved high performance values, with a classification accuracy of 94.77%, sensitivity of 92.23%, and specificity of 95.91%.

Recently, in 2022, authors in [10] introduced a novel concept of 'sentiment scope' within social network analysis, presenting a multi-dimensional framework to quantify users' influence on public sentiment over time and space, tested on a Reddit dataset. One year later, in 2023, the authors extended their work and proposed a scalable framework for analyzing sentiment dynamics among users and communities on social media, focusing on community influence, sentiment evolution, and inter-community interactions. Their approach has been empirically validated through extensive experiments [11].

Despite the progress made in the field, there are still limitations in these studies. The scope of these studies is limited to Twitter data, and the focus is on a limited number of languages and topics. Additionally, the methods used in these studies may not be able to accurately capture the full range of emotions and sentiments expressed in social media posts. Therefore, further research is needed to expand the scope and improve the accuracy of sentiment analysis on social media platforms.

### 2.2. Sentimental Analysis Using Arabic Language

Sentiment analysis using Arabic text has been a topic of interest in recent years. This is due to the increasing use of social media platforms and the growing amount of Arabic text data available. Sentiment analysis in Arabic text has been applied to a variety of domains such as politics [12], e-commerce [13], and customer reviews [14].

One of the main challenges in sentiment analysis of Arabic text is the lack of resources and lexicons for the Arabic language. Additionally, the complexity of the Arabic language and its use of diacritics and different forms of writing (e.g., classical, modern standard) make it difficult to accurately process and analyze the text.

Several studies have attempted to address these challenges by developing methods for preprocessing and normalizing Arabic text [15]. Other studies have proposed the use of machine learning techniques such as support vector machines and decision trees to classify the sentiment of Arabic text [16].

Despite the progress made in the field, there are still limitations in these studies. The methods used in these studies may not be able to accurately capture the full range of sentiments and emotions expressed in Arabic text. Additionally, the lack of resources and lexicons for the Arabic language is still a major challenge. Therefore, further research is needed to improve the accuracy of sentiment analysis of Arabic text and to develop resources for the Arabic

### 2.3. Sentimental Analysis in Education

The compilation of related work in sentiment analysis within the context of education reflects a trajectory of research focused on understanding and enhancing student experiences. Zhou et al. [17] conducted a study analyzing public sentiment towards online education in China through Sina Weibo microblogs during the pre-pandemic, amid-pandemic, and post-pandemic phases. This research provided insights into the evolution of public opinion on online learning, particularly during the critical times of the COVID-19 outbreak, highlighting the shift towards digital platforms in education. Building upon this context, Toccouglu et al. [18] applied sentiment analysis to student evaluations of online courses, utilizing machine learning to categorize feedback. The study underscored the utility of sentiment analysis in discerning student perceptions, which could be instrumental in improving online education. However, it also pointed out the subjective nature of sentiment interpretation and called for further validation of these analytical methods within educational settings. In a similar vein, the work of Nikolic et al. [19] and Mohiudddin et al. [20] both explored the application of sentiment analysis to student reviews in higher education. Their findings advocate for the use of sentiment analysis as a means to grasp and enhance student satisfaction and educational quality. They, too, noted the necessity for more refined analytical tools capable of capturing a wider spectrum of student emotions and sentiments, and a call for additional research to establish the robustness of sentiment analysis in educational evaluations was made. Lastly, D'Souza et al. [21] also employed machine

learning techniques for sentiment analysis of student feedback, reinforcing the potential of this approach in educational improvement. Like the other studies, it acknowledged the limitations in current methodologies and supported the need for advancements in sentiment analysis to fully interpret the complexities of student feedback. These studies collectively illustrate a concerted effort to apply sentiment analysis in educational settings, consistently indicating its value and also the ongoing challenges that need to be addressed. They highlight the transformative potential of sentiment analysis in capturing student perceptions, albeit with an understanding that methodological enhancements are essential for realizing its full potential.

*2.4. SVM for Sentimental Analysis*

The impact of machine learning [22,23] on the education system and how it is affecting human learners and learning is discussed recently in [24]. the authors compared deep learning in computers and humans and highlight the importance of explainability and accountability in machine learning systems. The article suggested that students should have an understanding of machine learning systems and opportunities to adapt and create them. It recommends changes to school curricula to address these issues and gives recommendations for teachers, students, policymakers, developers, and researchers. The analysis of the literature review on SVM for sentimental analysis in education reveals that there is no specific focus on the use of SVM for analyzing real-life data of Arabic review texts collected from students in higher education. However, there are several research articles that have used SVM for sentimental analysis in other domains and languages. One example is [25] a review and comparative analysis over social media is provided. The authors highlighted the use of SVM to classify sentiment in social media posts written in English. The authors highlighted that SVM performed well in comparison to other machine learning algorithms such as Naive Bayes and K-NN. Another example has used SVM to classify sentiment in user reviews written in English. The authors found that SVM performed well in comparison to other machine learning algorithms such as Naive Bayes, K-NN and Decision Tree [26]. In [27] researchers presented state-of-the-art approaches for aspect-based sentiment analysis (ABSA) of Arabic hotel reviews, using supervised machine learning techniques, including deep recurrent neural networks (RNN) and SVM. The study used various features, such as lexical, word, syntactic, morphological, and semantic, to train and evaluate the models. Results showed that the SVM approach outperformed the deep RNN approach for the tasks of aspect category identification, aspect opinion target expression extraction, and aspect sentiment polarity identification. However, the deep RNN approach had a faster execution time, particularly for the second task. It's worth noting that there are several papers that use SVM on Arabic sentiment analysis but the data are not always real-life data which is an Arabic review text collected from students in higher education. For example, the authors in [28] focused on analyzing sentiment in e-learning using traditional machine learning algorithms, specifically SVM and Naive Bayes. They used a dataset of 2000 tweets related to King Abdul-Aziz University to train and evaluate their models. For example in [29], the authors proposed Arabic Language Sentiment Analysis (ALSA). ALSA is a challenging task due to the complex nature of the Arabic language. The language has multiple levels such as phonetics, morphology, syntax, lexicology, semantics, and figurative nature. These levels have positive and negative connotations and meanings that need to be considered in the analysis. ALSA has been less studied compared to the analysis of opinions and feelings in English and Indo-European languages. Therefore, there is a need for a comprehensive and full proposal of a strategy for ALSA. This includes analyzing opinions and feelings at all levels of language, in addition to the importance of building an annotated corpus that helps to understand an Arabic sentence from the level of phonetics to the rhetorical and metonymy levels. In conclusion, while there is a lack of literature specifically focused on the use of SVM for sentimental analysis of real-life data of Arabic review texts collected from students in higher education, there are several research articles that have used SVM for sentimental analysis in other domains

and languages. Therefore, it is possible that SVM could be used for sentimental analysis of Arabic review texts collected from students in higher education, but more research is needed to confirm this. In addition, the proposed method seeks to advance the sentiment analysis field by offering superior analyzing accuracy beyond the achievements of previous studies, encompassing a broader spectrum of sentiments beyond the basic emotional ranges typically captured, and extending the analysis to a wider variety of textual data, notably addressing the challenges posed by the Arabic language's complexity. The SVM-SAA-SCR technique provides tangible improvements over the limitations highlighted in earlier research, including better performance metrics, a more comprehensive sentiment detection, and adaptability to diverse and complex Arabic language datasets.

## 3. Contributions

To the best of our knowledge, there is currently no attempt to develop a sentimental analysis system for high education students using Arabic text. This is an important gap in the literature as there is a need to understand the perceptions and opinions of students in Saudi Arabia Universities regarding their education beyond COVID-19. Developing such a system would provide valuable insights into the challenges and obstacles faced by Arab students during online learning and help to improve their educational experience. The main contributions of this study include:

1. The proposed SVM Sentimental Analysis for Arabic Students' Review called SVM-SAA-SC algorithm for students' Arabic course reviews is a general framework that involves collecting student reviews, preprocessing them, and using a machine learning model to classify them as positive, negative, or neutral.
2. The suggested technique for preprocessing and classifying reviews includes steps such as collecting data, removing irrelevant information, tokenizing, removing stop words, stemming or lemmatization, and using pre-trained sentiment analysis models.
3. To train the classifier, SVM algorithm is used, and performance is evaluated using metrics such as accuracy, precision, and recall.
4. Fine-tuning is done by adjusting parameters such as kernel type and regularization strength to optimize performance.
5. The use of a real dataset provided by the deanship of quality at PSAU, which is ranked 801–900 globally and 6th in Saudi Arabia according to the 2022 Shanghai Rankings.
6. The dataset contains students' opinions on various aspects of their education, including courses, faculties, assignments, projects, and exams.
7. To validate the suggested SVM-SAA-SC algorithm, a comparison with a sentiment analysis tool called CAMeLBERT Dialectal Arabic model (CAMeLBERT-DA SA Model) [30] is adopted. CAMeLBERT represents an interplay of variant, size, and task type in Arabic pre-trained language models.
8. The use of textual comments instead of grading-based feedback to reveal the exact sentiment of students.

The proposed model provides insights about education at PSAU and contributes to improving the quality of teaching at PSAU.

## 4. Dataset

The dataset used in this study was collected from PSAU, specifically from the College of Computer Engineering and Sciences (CCES) via student course evaluation surveys post-COVID-19 (refer to Figure 1). The data was gathered by the quality unit of CCES, resulting in a total of 384 records and 30 features from 30 different courses spreading across 66 sections. Figure 2 provides a sample of the dataset used. The features include both categorical and quantitative variables, as well as textual comments. Table 1 provide a translation of the Arabic comments provided in Figure 2. Table 2 provides an overview of the extracted features and their descriptions. We highlight that the proportion of class labels in our dataset refers to the distribution of different categories or classifications within the data. For instance, in a sentiment analysis task, the class labels is maily considered to



be "positive", "negative", and "neutral", each corresponding to the sentiment expressed in a piece of text. The proportion of these labels would then indicate how many of the entries in the dataset are positive, negative, or neutral, typically represented as a percentage or a count. The proportions of each class within the dataset are as following:

- Positive Class: Significantly more than half of the data, 57.1%.
- Neutral Class: Roughly around half the number of Positive samples, 28.6%.
- Negative Class: Roughly around a quarter of the number of Positive samples, 14.3%.

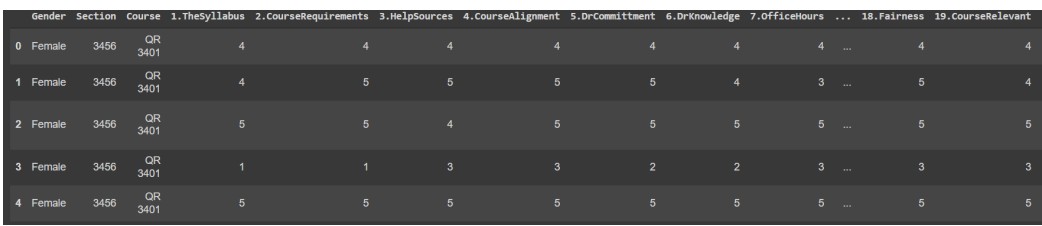

**Figure 1.** Sample of students' course evaluation survey.

| reviews | sentiment |
|---|---|
| الدكتورة الهام كريري ساعدتنا في تبسيط المنهج ومادتها كانت اسهل وافضل المواد عندي هذا الترم ومراعية جدا لظروفنا الله يجزاها خير هي كانت سبب في تسهيل وتبسيط المنهج لنا | positive |
| الدكتورة كانت فعلا ممتازه ومراعيه وشرحها جميل ومفهوم شكرا لها | positive |
| الدكتوره | neutral' |
| الدكتوره العسسسل دكتوره سناء وشرحها | neutral' |
| الدكتووووره جبر خاطر واهي محببتني في الماده وباذن الله لما اتخرج اشتغل بقواعد بيانات | positive |
| السلايدات جيده لكن تحتاج لتطوير الافكار بعض النقاط مبهمه ولا يوجد فيها شرح كافي | neutral' |
| السلايدات غير واضحة | negative |
| السلايدات هي ممتازة لكن لو حصل تطوير فيها بحيث انه يكون فيه شرح للاكواد بشكل نظري كا مثال و بعدها ننتقل للعملي | neutral' |
| الشرح | neutral' |
| الصعوبه في الاختبارات | neutral' |
| الضغط ف اللابات والواجبات | negative |
| Business Architecture العمل اكثر علي تحقيق شهادة | positive |
| العمل الجماعي وانه شي جديد نتعلمه الي هو البرمجه | positive |
| العملي | neutral' |
| الفاءدة | negative |
| الفكرة عن تنظيم المعلومات | neutral' |
| الكثير من الشباتر | negative |
| الكثير من المواد النظريه اعتمد فيها علي الحفظ اكثر من الفهم بس في الجزء النظري لهذا المقرر استطعت فهمه بشكل تام و كان ممتع | positive |
| الكويزات الي تسبق الدرس الي لم ناخذه بعد مستوي صعوبة بعض الاسءلة لا يناسب الطلاب بحكم عدم تخصصهم في المجال | negative |
| الكويزات بداية المحاضرة قبل ماناخذها وهذا الشيء احس يخوف | negative |

**Figure 2.** Sample of students' comments.

**Table 1.** Translation of students comments illustrated in Figure 2.

| Review Translated | Sentiment |
| --- | --- |
| Dr. Elham has faciltated the course content and was very supportive.. | Positive |
| The professor explanation was very good and understandable. | Positive |
| The professor. | Neutral |
| The professor is amazing. | Neutral |
| I like the professor. When I graduate, I will be a database administrator. | Positive |
| The slides are fair but need improvement. | Neutral |
| Slides are not clear. | Negative |
| Slides are good. I think thet deserve to contain more details. | Neutral |
| Explanation. | Neutral |
| Exams are quite difficult. | Neutral |
| We are pressed by the labs and homeworks. | Negative |
| More effort to reach a certificate in Business Architecture. | Positive |
| Team work and I think I found great what we are learning in programming. | Negative |
| The variety of tasks and the presence of a real challenge in the work is what makes the experience unique and valuable. | Positive |
| Practical. | Neutral |
| Benefit. | Negative |
| Idea on organizing information. | Neutral |
| Lot of chapters. | Negative |
| Despite the quantity of theoretical courses that require remembering more than understanding, I was able to understand clearly and it was funny. | Positive |
| Quizzes are hard. | Negative |
| I am quite scared from quizzes that we take at the beginning of lectures. | Negative |

**Table 2.** Features extracted from the datase.

| Feature | Description | Data Type |
| --- | --- | --- |
| Gender | Gender of the student | Categorical |
| Section | Course section number | Categorical |
| Course | Course The Course code | Categorical |
| 1. TheSyllabus | The course syllabus | Numerical |
| 2. CourseRequirements | The requirements for success in the course (including the duties to be assessed and the criteria for assessment) were clear to me | Numerical |
| 3. HelpSources | Helping Sources | Numerical |
| 4. CourseAlignment | The implementation of the course and the things I was asked to do aligned with syllabus | Numerical |
| 5. DrCommittment | The faculty member was committed to giving the course completely (such as: starting lectures on time, presence of the faculty member, good preparation of teaching aids, and so on) | Numerical |
| 6. DrKnowledge | The knowledge of instructor about the course | Numerical |
| 7. OfficeHours | The instructor was available to help during office hours | Numerical |
| 8. DrEnthusiasm | The instructor was enthusiastic about what he/she taught | Numerical |
| 9. DrMotivation | The instructor was interested in my progress and was a supporter to me | Numerical |
| 10. CourseNovelty | Everything presented in the course was modern and useful (reading texts, summaries, references, and the like) | Numerical |
| 11. CourseReferences | The sources I needed in this course were available whenever I needed them | Numerical |
| 12. UseOfTechnology | There was effective use of technology to support my learning in this course | Numerical |
| 13. EncouragementToAsk | I found encouragement to ask questions and develop my own ideas in this course | Numerical |
| 14. EncouragementToBest | In this course I was encouraged to do my best | Numerical |
| 15. KnowledgeDevelopment | The things that were asked of me in this course (classroom activities, labs, and so on) helped in developing my knowledge and skills that the course aims to teach | Numerical |
| 16. WorkHoursConsistency | The amount of work in this course was proportional to the number of credit hours allocated to the course | Numerical |
| 17. FeedbackTime | received the grades for assignments and tests in this course within a reasonable time | Numerical |
| 18. Fairness | The correction of my homework and exams was fair and appropriate | Numerical |
| 19. CourseRelevant | The link between this course and the other courses in the program (department) was explained to me | Numerical |
| 20. CourseImportance | What I learned in this course is important and will benefit me in the future | Numerical |
| 21. ProblemSolving | This course helped me to improve my ability to think and solve problems instead of just memorizing information | Numerical |
| 22. TeamWork | This course helped me to improve my skills in working as a team | Numerical |
| 23. Communication | This course helped me to improve my ability to communicate effectively | Numerical |
| 24. CourseSatisfaction | I am generally satisfied with the level of quality of this course | Numerical |
| 25. Strengths | What did you especially like about this course? | Textual |
| 26. Recommendations | What did you not especially like about this course? | Textual |
| 27. ImprovementsArea | What suggestions would you make to improve this course? | Textual |

### 4.1. Dataset Cleaning

Data preprocessing is the method of changing raw data into an understandable format. It is additionally a critical step in any data science project [31]. The quality of the data must be checked before applying machine learning techniques to increase results credibility and reliability. During this phase, we prepare collected data for further processing. This involves four steps, which are described in the following subsections.

### 4.2. Removing Straightlining

Straightlining is when a respondent chooses the same answer option over and over again along all questions (e.g., the last answer option). Straightliners are often speeders as well, as they race through the survey by answering each question with little to no thought. In our survey we found 60 straightlining responses which represents 15.625% of the dataset. These responses have a high impact and could lead to misleading results. After this step we ended up with 324 observations. We performed several preprocessing steps on the datasets. The objective of these measures was to remove elements of the text that could negatively affect the quality of the results.First, we removed diacritical marks from the text. This is a common text preprocessing step in Arabic NLP. To do this, we simply specified a list of these characters and removed them from both datasets. The second preprocessing step was to normalize several Arabic letters which could be written in different manners but have the same meaning. The reason behind this normalization is because we had observed that the data probably (آ) because consists of data collected from a group of students, which tend to be written informally. For example, all instances of We also noticed that some texts included repeated (ي) were changed to (ى) and all instances of

(أإآ) were changed to (ا). In text preprocessing for NLP, normalizing repetitive characters and diacritical marks, as commonly done in Arabic NLP, has influenced our model's generalization capabilities. This process reduces noise, ensuring consistency across word variants and decreasing the vocabulary size, which can streamline training and potentially improve model performance on new, unseen data by reducing out-of-vocabulary errors. However, such normalization can also remove nuanced information, such as emotional intensity indicated by character elongation, which might impact the model's sensitivity to the subtleties of language. Therefore, the benefits of normalization, which include helping the model focus on relevant features and achieve better generalization, must be weighed against the potential loss of specific linguistic details, especially when applying the model in varying contexts that may demand recognition of formal versus informal text structures.

1.  Tokenization
    To identify the sentiments and extract emotions of sentences, Students' reviews (e.g., comments) are split into words, or tokens, this task is completed using the tokenize function in the python package natural language toolkit (NLTK).
2.  Normalization
    We also detect that some comments included repeated characters. In colloquial written speech, some letters are frequently repeated in order to focus on certain emotions. To normalize how this type of emotive word was written, we removed repeated letters and replaced all instances with one only. For example, the word رااااائع was changed to رائع.
3.  Removal of irrelevant content
    Punctuation, stop words, symbols, and prepositions, which are irrelevant for Sentiment analysis, are removed to improve the results, response time and effectiveness.
4.  Stemming
    Stemming is the most daunting task due to the need to review lexicons. To further facilitate word matching, words in student comments are converted to their root word. For example, المحاضرات, المحاضر are all converted to محاضرة.

*4.3. EDA*

Data exploration, or Exploratory Data Analysis (EDA), is the process of analyzing and understanding the patterns, outliers, and relationships in a dataset. It is a crucial step in any data science project, as it allows us to gain a deeper understanding of the data and make informed hypotheses. One powerful tool for EDA is the pandas profiling library, which is an open-source Python package that allows us to perform EDA quickly and easily with just a few lines of code. Additionally, it generates web reports that can be shared with others, making it an ideal tool for presenting our findings. With pandas profiling, we can save time and effort by easily visualizing and understanding the distribution of each variable in the dataset. It creates a comprehensive report that includes summary statistics (Table 3), tables, and visualizations, making it a valuable tool for data exploration and understanding. Additionally, the report also includes information about missing values, such as the Count, Heatmap, and Dendrogram. The Count chart, shown in Figure 3, illustrates the number of missing values in each feature and the count of it. The Dendrogram, shown in Figure 4, displays the missing values in a hierarchical format. It's important to note that all missing values are textual in nature, as filling in textual data in the survey is optional. To further understand the relationships between variables (features) in our dataset, we used a correlation matrix. The square matrix is based on the Spearman correlation coefficient, represented by $\rho$. The correlation value describes the interaction strength between two features. The range of correlation varies from +1 to −1. A positive value indicates a positive association between two features, while a value of 0 means that there is no association between these two features, or the relationship is not linear. Thus, they are independent from other features. The correlation matrix provides a clear picture of the association between features. Figure 5 illustrates the matrix for 24 selected features. Additionally, we created a scatter plot using the Seaborn library to represent the pairwise correlations between the various variables. This is useful in providing a simple summary of how the data is spread and whether it includes outliers. Figure 6 illustrates the scatter plot to display the correlation between the highest positive correlation based on Figure 7 where the red shadow in this figure represents the confidence interval for the regression line, indicating the uncertainty about the line's true position based on the variability of the data points. This association shows that the teamwork and communication features have a strong correlation coefficient of 0.83.

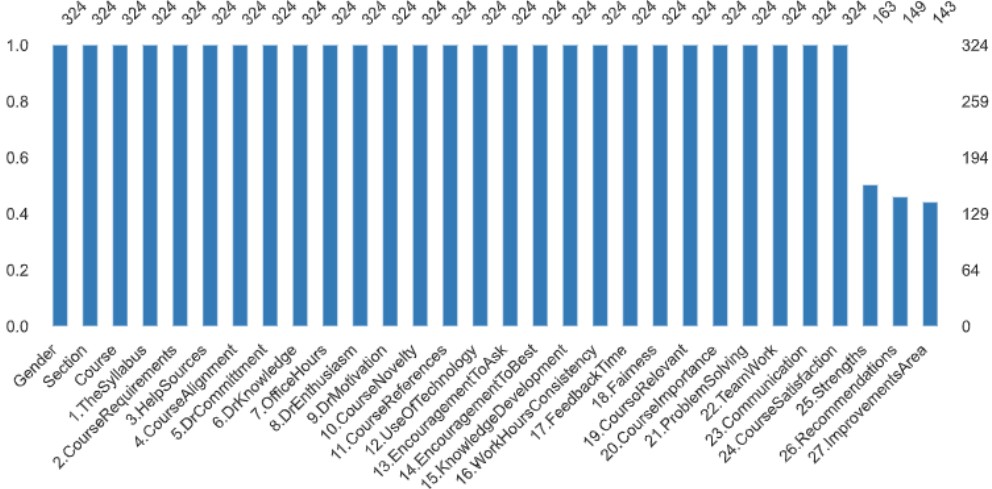

**Figure 3.** Missing Values Count.

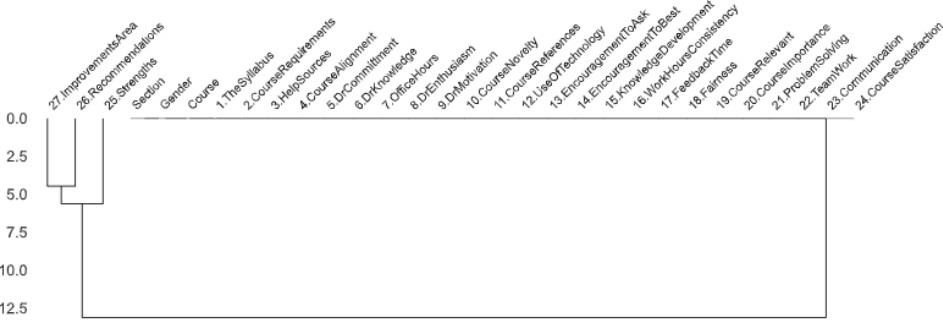

The dendrogram allows you to more fully correlate variable completion, revealing trends deeper than the pairwise ones visible in the correlation heatmap.

**Figure 4.** Missing Values, Dendrogram.

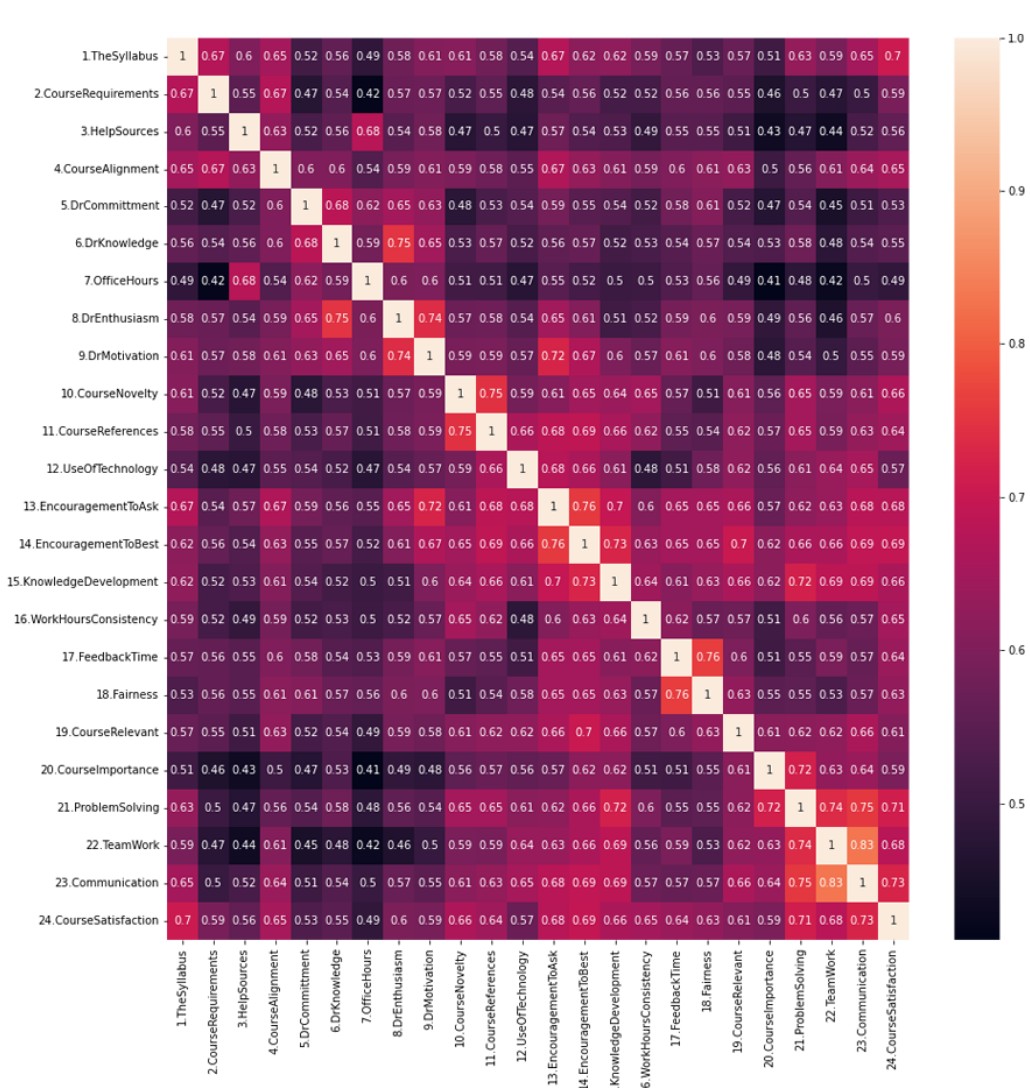

**Figure 5.** Correlation Matrix.

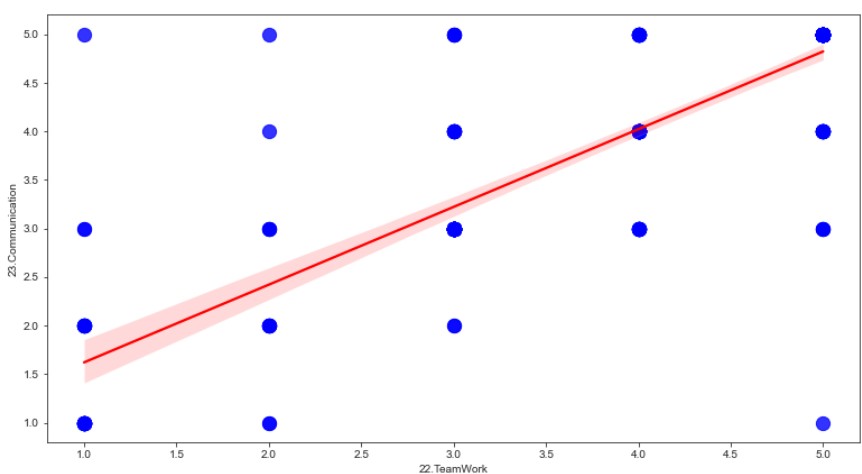

**Figure 6.** Correlation between Teamwork and Communication.

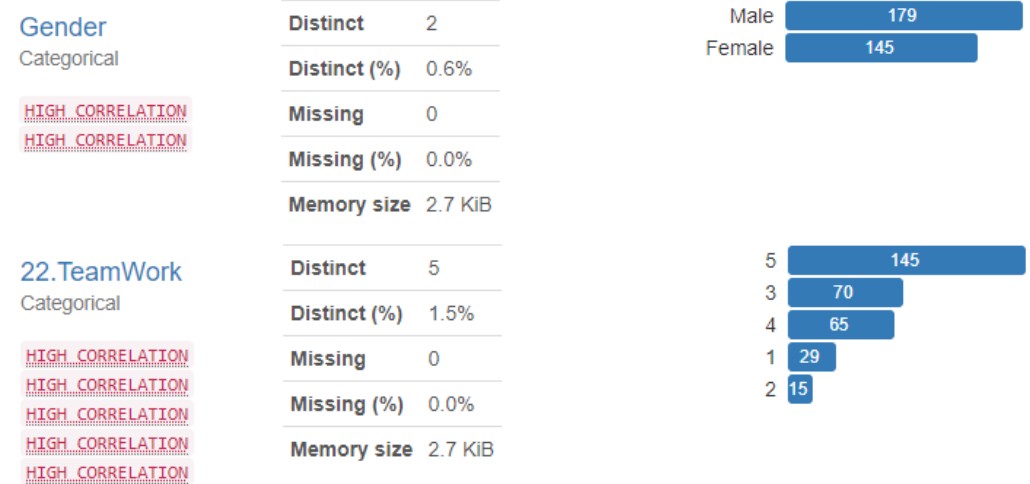

**Figure 7.** Variables Overview.

**Table 3.** Dataset Statistics.

| | |
|---|---|
| Number of variables | 30 |
| Number of observations | 324 |
| Missing cells | 517 |
| Missing cells (%) | 5.3% |
| Duplicate rows | 0 |
| Duplicate rows (%) | 0.0% |
| Total size in memory | 76.1 KiB |
| Average record size in memory | 340.4 B |

## 5. Methods

*5.1. SVM-SAA-SCR*

To develop our system, we used a C-Support Vector Classification (C-SVC) approach, which is widely used for classification problems and allows for tuning of the kernel type

and regularization strength to find the best performing model on the training data [32]. The proposed sentimental analysis algorithm for students' Arabic course reviews is as follows:

1. Collect a dataset of student reviews for an Arabic course.
2. Preprocess the reviews by removing any irrelevant information such as special characters and numbers.
3. Tokenize the reviews to create a list of words for each review.
4. Create a sentiment dictionary or use pre-trained sentiment lexicon to classify each review as positive, negative, or neutral.
5. Use a machine learning model to train on the preprocessed and classified reviews.
6. Use the trained model to classify new reviews.
7. Analyze the results to identify patterns and areas for improvement in the Arabic course based on the sentiment classification of the reviews.
8. Repeat the process with updated reviews and continue to make improvements to the course based on the analysis of the results.

The suggested algorithm is a general framework and different techniques can be used to preprocess and classify the reviews. Additionally, the algorithm can be improved by using additional techniques such as removing stop words, using stemming or lemmatization, and using advanced machine learning models.

### 5.1.1. SVM-SAA-SCR Preprocessing and Classification

The proposed technique for preprocessing and classifying students' Arabic course reviews includes the following steps:

1. Collect a dataset of students' Arabic course reviews from various sources, such as online forums, social media, and surveys.
2. Preprocess the reviews by removing any irrelevant information, such as URLs, usernames, and special characters.
3. Tokenize the reviews by splitting them into individual words or phrases.
4. Remove stop words, such as "and" and "the", which do not carry much meaning in the analysis.
5. Perform stemming or lemmatization to reduce words to their base form and improve consistency in the dataset.
6. Use pre-trained sentiment analysis models, such as SentimentIntensityAnalyzer or TextBlob, to classify each review as positive, negative, or neutral.
7. Use the resulting sentiment scores to create visualizations, such as bar charts or pie charts, to show the distribution of sentiments among the reviews.
8. Analyze the results to identify patterns and trends in the students' sentiments towards the course.
9. Use the insights gained from the analysis to improve the educational process and address areas of improvement.

Please note that this is a general approach and it may require some adjustments to suit the specific requirements of the task. Also, you may need to preprocess the data more deeply to handle the Arabic language specifics such as diacritics, or even use specialized libraries for Arabic text processing.

### 5.1.2. Model Training Based SVM-SAA-SCR

To train the SVM-SAA-SCR, we used the course reviews of PSAU students provided by the deanship of quality in the College of Computer Engineering and Sciences. Then the following steps are adopted:

1. Collect and preprocess the reviews by removing any irrelevant information such as diacritics, URLs, and names.
2. Split the dataset into training and testing sets.
3. Convert the text data into numerical features using techniques such as bag-of-words or TF-IDF.

4. Train an SVM classifier using the training dataset and the numerical features as input.
5. Test the classifier on the testing dataset and evaluate its performance using metrics such as accuracy, precision, and recall.
6. Fine-tune the model by adjusting parameters such as the kernel type and regularization strength to optimize performance.
7. Use the trained and fine-tuned classifier to classify new reviews and predict their sentiment.

### 5.1.3. Evaluation of SVM-SAA-SCR Performance

The performance of the suggested SVM-SAA-SCR can be evaluated using several metrics, such as accuracy, precision, and recall. Here is one possible approach to evaluate the performance of the SVM-SAA-SCR:

1. Split the dataset into a training set and a test set.
2. Train the SVM model on the training set.
3. Use the trained model to predict the labels of the samples in the test set.
4. Compare the predicted labels to the true labels of the test set samples.
5. Compute the accuracy, precision, and recall of the model using the following formulas:
    - Accuracy = (True Positives + True Negatives)/Total Samples
    - Precision = True Positives/(True Positives + False Positives)
    - Recall = True Positives/(True Positives + False Negatives)
6. Interpret the results and make any necessary adjustments to the model or the data preprocessing steps.

We note that accuracy is the percentage of correctly classified samples, precision is the percentage of correctly classified positive samples among all the positive samples, recall is the percentage of correctly classified positive samples among all the actual positive samples. The suggested SVM-SAA-SCR has the ability to classify sentiments using a small training set (PSAU dataset). It has also shown strong competition in achieving a high accuracy of 84.7%. The detailed performance results based on the PSAU dataset are detailed in Table 4 and Figure 8

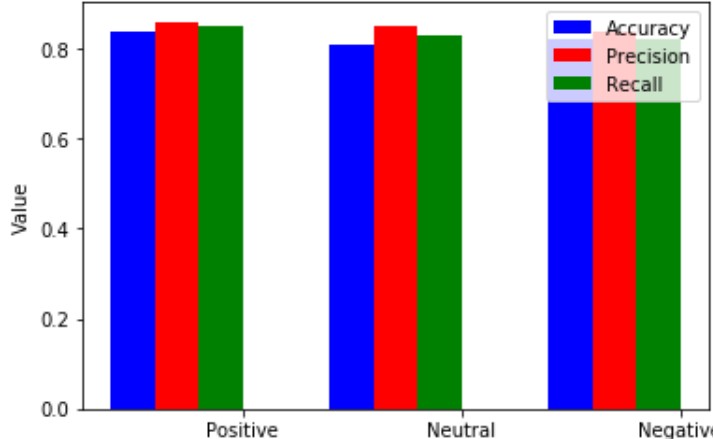

**Figure 8.** SVM-SAA-SCR performance on on PSAU dataset.

**Table 4.** SVM-SAA-SCR performance on PSAU Dataset.

| Class | Accuracy | Precision | Recall |
| --- | --- | --- | --- |
| Positive | 0.84 | 0.86 | 0.85 |
| Neutral | 0.81 | 0.85 | 0.83 |
| Negative | 0.82 | 0.84 | 0.82 |

5.1.4. Tuning Hyper-Parameters for SVM-SAA-SCR

To tune the hyper-parameters of SVM-SAA-SCR, we adopted the way suggested by Louati et al., (2022) [31], which could be summarized as follows:

1. Begin by initializing the model with a default kernel type (e.g., radial basis function kernel) and regularization strength (e.g., C = 1).
2. Use cross-validation techniques to evaluate the performance of the model on the training data, using metrics such as accuracy, precision, and recall.
3. Experiment with different kernel types (e.g., linear, polynomial, sigmoid) and regularization strengths (e.g., C = 0.1, C = 10) to find the combination that gives the best performance on the training data.
4. Once the optimal kernel type and regularization strength have been identified, use these parameters to train the model on the entire training set.
5. Evaluate the performance of the optimized model on the test set using the same metrics as before.
6. If the performance is not satisfactory, fine-tune the model further by adjusting other parameters such as the degree of the polynomial kernel or the gamma parameter for the radial basis function kernel.
7. Repeat the process of evaluating and fine-tuning the model until the desired level of performance is achieved.

To avoid overfitting L1 or L2 regularizations are adopted, which add a penalty term to the loss function to discourage the model from assigning too much weight to any one feature. Additionally, an early stopping technique is used to prevent the model from memorizing the training data. To prevent underfitting, we adopted data augmentation to expand the size of the training dataset.

*5.2. CAMeLBERT-DA SA Model*

This CAMeLBERT-DA SA Model is implemented based on the following datasets:

1. The Arabic Speech-Act and Sentiment Corpus of Tweets (ArSAS), where 23,327 tweets have been used [33];
2. The Arabic Sentiment Tweets Dataset (ASTD) [34];
3. SemEval-2017 task 4-A benchmark dataset [35];
4. The Multi-Topic Corpus for Target-based Sentiment Analysis in Arabic Levantine Tweets (ArSenTD-Lev) [36].

These datasets were combined and preprocessed in a similar way to what was done in [37,38]. The preprocessing steps included removing diacritics, URLs, and Twitter usernames from all the tweets.

The experimental setup adopted by [30] for CAMeLBERT-DA SA is summarized as follows:

1. Build three pre-trained language models across three variants of Arabic: Modern Standard Arabic (MSA), dialectal Arabic, and classical Arabic.
2. Build a fourth language model pre-trained on a mix of the three variants.
3. Build additional models pre-trained on a scaled-down set of the MSA variant to examine the importance of pre-training data size.
4. Fine-tune the models on five NLP tasks spanning 12 datasets.
5. Compare the performance of the models to each other and to eight publicly available models.
6. Analyze the results to identify the importance of variant proximity of pre-training data to fine-tuning data and the impact of pre-training data size.
7. Define an optimized system selection model for the studied tasks based on the insights gained from the experiments.
8. Make the created models and fine-tuning code publicly available.

## 6. Results

Figures 9 and 10 provide an illustration of the sentiment polarity for a total of 1870 reviews. These reviews have been classified into three categories: positive, negative, and neutral. Table 5 represents a comparison between SVM-SAA-SCR and CAMeLBERT-DA SA based on these 1870 students' reviews. Figure 9 shows that the majority of reviews are positive, with roughly a quarter of the reviews being neutral. This information can be used to identify areas of improvement in the educational process. Similarly, Figure 10 illustrates the sentiment polarity for 1870 reviews, with the majority of these reviews being positive. Again, this information can be used to pinpoint areas of improvement in the educational process. In summary, the CAMeLBERT-DA SA Model shows that 70.48% of the reviews are classified as positive, while the SVM-SAA-SCR shows that 69.62% of the reviews are classified as positive. Additionally, the CAMeLBERT-DA SA Model shows that 22.03% of the reviews are classified as neutral, while the SVM-SAA-SCR shows that 20.74% of the reviews are classified as neutral. Lastly, the CAMeLBERT-DA SA Model shows that 7.48% of the reviews are classified as negative, while the SVM-SAA-SCR shows that 9.62% of the reviews are classified as negative.

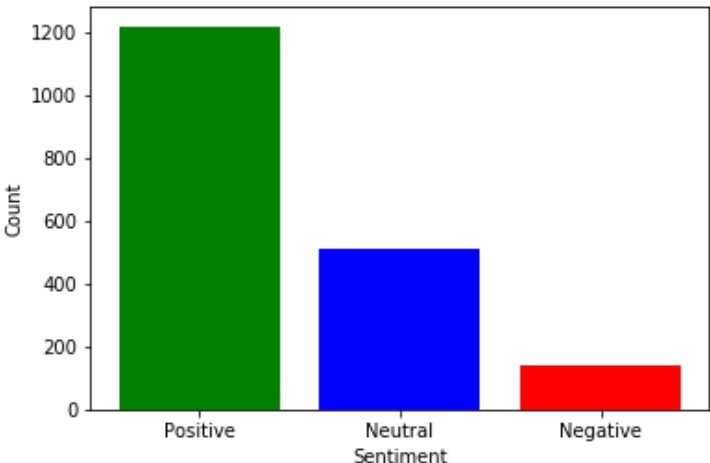

**Figure 9.** CAMeLBERT-DA SA Model result.

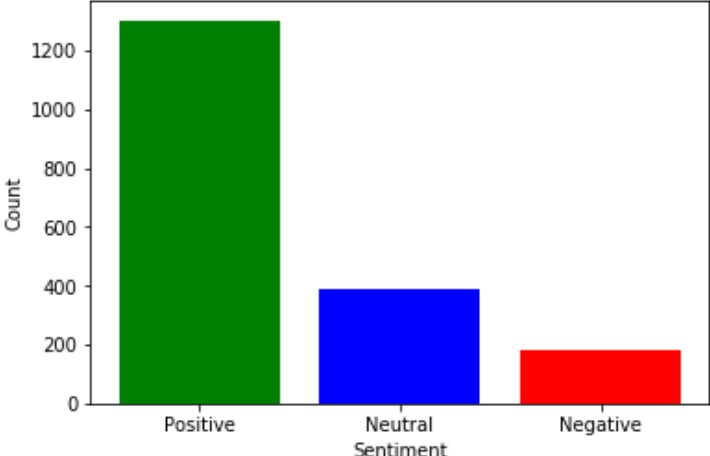

**Figure 10.** SVM-SAA-SCR result.

**Table 5.** SVM-SAA-SCR vs. CAMeLBERT-DA SA based on PSAU dataset of 1870 students' reviews.

| Class (Count) | SVM-SAA-SCR | CAMeLBERT-DA SA |
|---|---|---|
| Positive | 1302 | 1318 |
| Neutral | 388 | 412 |
| Negative | 180 | 140 |

## 7. Discussion

The proposed sentimental analysis system for high education students using Arabic text can provide valuable insights into the perceptions and opinions of students at PSAU. The analysis of student reviews, classified as positive, negative, or neutral, can help identify areas of improvement in the educational process. This information can be used to make necessary adjustments to the curriculum and teaching methods to enhance the educational experience for students. Additionally, the system can help identify specific issues that students may be facing and provide targeted solutions to address these issues. By providing a more detailed understanding of student perceptions and opinions, the proposed system can help to improve the overall quality of education at Prince Sattam University. Furthermore, the system can also help in improving the quality of teaching by providing teachers with feedback on their performance, which can help them to make necessary adjustments and improve their teaching methods.

Compared to the literature, our work enhanced sentiment analysis literature by providing improved analytical precision, surpassing prior research outcomes. It captures an extensive array of sentiments, going beyond traditional emotional categories, and broadens the scope of analysis to include a diverse range of text data, with a particular focus on overcoming the intricacies of the Arabic language. The SVM-SAA-SCR technique offers marked advancements, such as enhanced performance indicators, expanded sentiment identification, and increased flexibility for various complex Arabic datasets. Finally, we draw the reader's attention that this work sought to validate the suggested system with a well-known tool such as CAMeLBERT through a comparison based on accuracy and results similarity. This comparison has shown a very slight difference between both systems and promising results for SVM-SAA-SCR.

## 8. Conclusions

The study conducted an in-depth development of a sentimental analysis system tailored for higher education students using Arabic text, marking a significant stride in the relevant academic field. Our SVM-SAA-SCR algorithm, applied to the real dataset from Prince Sattam bin Abdulaziz University (PSAU), was set against the backdrop of the sophisticated CAMeLBERT Dialectal Arabic model, allowing us to evaluate student sentiments about their educational experiences in a post-COVID-19 context. Results highlighted a predominantly positive outlook, with both models revealing over 69% positive feedback. Furthermore, when we delve into the performance metrics of the SVM-SAA-SCR model, we find that it not only performs sentiment classification with a limited training set but also exhibits robustness with an impressive accuracy rate of 84.7%. The presence of negative and neutral sentiments, albeit in a smaller proportion, pinpoints areas that are ripe for improvement within the educational framework. The deployment of sentiment analysis thus emerges as an essential tool for educators and administrators, keen on optimizing the educational milieu based on student feedback. In summary, this investigation underscores the potential of advanced sentiment analysis to extract actionable insights from student feedback, thereby enhancing the overall educational quality and experience at PSAU. It serves as a testament to the power of machine learning in transforming qualitative data into a catalyst for educational innovation.

**Author Contributions:** Methodology, A.L.; Software, A.L., F.A. and A.A.; Validation, A.L.; Data Acquisition: A.L.; Writing—original draft preparation, A.L., F.A.; Review and editing, A.L., H.L.; Supervision, A.L.; Project administration, E.K. All authors have read and agreed to the published version of the manuscript.

**Funding:** Prince Sattam bin Abdulaziz University project number (PSAU/2023/R/1444).

**Institutional Review Board Statement:** Not applicable.

**Informed Consent Statement:** Not applicable.

**Data Availability Statement:** The data could be provided on reasonable request from the corresponding author.

**Acknowledgments:** The authors thank the Deanship of Scientific Research at Prince Sattam bin Abdulaziz University (PSAU) for funding this work via project number (PSAU/2023/R/1444). In addition, the authors thank Luay Assidmi (l.assidmi@psau.edu.sa) the ex-dean of quality in the College of Computer Engineering and Sciences, PSAU for providing the dataset. Finally, The authors would like to acknowledge that this research work was partially financed by Kingdom University, Bahrain from the research grant number 2023-11-015.

**Conflicts of Interest:** The authors declare no conflict of interest.

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
