# Peer review of "Sentiment Analysis of Arabic Course Reviews of a Saudi University Using Support Vector Machine"

_applsci, doi:10.3390/app132312539_

Round 1
Reviewer 1 Report
Comments and Suggestions for Authors
abstract: The proposed method gives lower performance than the baseline.
Intro:
1. the background does not contain relevant references. As a result, the research problem to be solved is not clear.
2. Regarding the SVM method chosen to represent machine learning, there is no reference to support the reason for its selection as well
Dataset: describe the proportion of class labels in the dataset
Method: The proposed SVM-SAA-SCR method seems to be unified, but in actuality, each stage is like a normal text classification process.
Result:
The sentiment comparison shown, however, does not prove which method is superior, as the results of manually labeling the data are not given. This also applies to the use of accuracy measures in this research, it should also be based on the proportion of manual labeling results. to compare the best method, the author should use 2 measures such as accuracy and F1-measure, not based on the size of the each class sentiment.
conclusion: needs to be adjusted again with the results of the final analysis
Comments on the Quality of English Languageok
Author Response
We thank the reviewer for his pertinent comments.
A detailed letter is provided for reviewers addressing each comment. Detailed answers and rebuttals to these comments are provided, and the manner how the article was updated to accommodate them is shown. In the revised version, the modifications suggested by reviewers are highlighted in blue.

Reviewer 2 Report
Comments and Suggestions for Authors
The paper presented a SVM-based system to analyze the sentiments of higher education students’ course reviews in Arabic, and compared it to a baseline dialectal Arabic model CAMeLBERT to show its preferable performance.
The topic of the paper is interesting. My concerns are as follows.
In Section “Related Work”, the authors reviewed quite a few studies from related fields and provided respective comments on them. The limitations mentioned by the authors in these studies can be roughly classified into three aspects: inadequate analyzing accuracy, not covering full range of sentiments, and limited scope of textual data. In the last paragraph, the readers may expect to know the things specific to the proposed method, as well as the differences between this study and the related work, especially in the above-mentioned aspects.
In Section “Dataset”, a machine learning model was used to classify the samples as positive, negative or neutral. However, what are the proportions of the three classes of data? Accordingly, is there any hyperparameter that controls their contributions to the model in training? and how is it set?
The contents in Fig. 2 doesn’t make any sense to non-Arabic readers. An accompanying English translation for the sentences seems necessary.
Line 260, Page 6: The authors removed the repetitive characters and replaced them with single characters. Will the processing of data affect the generalization capability of the trained model? Please give some explanation on this point.
In Section “Methods”, a SVM-based model, called SVM-SAA-SCR is introduced. But the authors didn’t elaborate the model. What kind of SVM is employed, and is there any modification made to customize it to this kind of task?
In Section 5.2, the authors compared the proposed model with CAMeLBERT-DA SA. However, it seems that the two models didn’t adopt the same datasets for training.
In Section 6, The authors conducted a comparative experiment between CAMeLBERT-DA SA and the proposed model SVM-SAA-SCR. The results showed that the former classified 70.48% of the comments as positive, while the latter classified 69.62% as positive. The authors concluded that this proves the efficiency of SVM-SAA-SCR. Is this conclusion solely based on the lower proportion of the class “positive”? It is a bit confusing that the authors compared the two models using the classifying results instead of the values of evaluation metrics, as used in Table 3.
Comments on the Quality of English Language
Some expressive redundances can be reduced or removed, e.g.,
In “Abstract”, the last two sentences can be combined into one;
Line 225, “Model” in the parentheses can be removed;
Line 237, “Figure 2 in the paper …” -> “Figure 2”;
And typos, e.g.,
Line 237, “courses spread across …” -> “courses spreading across…”.
Author Response

(The authors gave the same response as above.)

Reviewer 3 Report
Comments and Suggestions for Authors
In this paper, the authors present a sentiment analysis system for high education students using Arabic text.
The topic of sentiment analysis is much studied in the literature; the main novelty proposed by the authors is its application to the context of Arabic text.
The study presented by the authors in the paper is interesting; although at technical level the novelties are few, at the level of analysis campaign and testing there is a wealth of experiments and results.
I have two main suggestions to the reviewers:
- The related work part on sentiment analysis should be enriched by including other recent papers on this topic. For example, the authors should mention among others, the following papers, "A framework for investigating the dynamics of user and community sentiments in a social platform" and "A Space-Time Framework for Sentiment Scope Analysis in Social Media."
- The Discussion section should be enriched with additional considerations regarding the originality of the results found compared to what is already in the literature.
Comments on the Quality of English LanguageThe English Language appears good
Author Response

(The authors gave the same response as above.)

Reviewer 4 Report
Comments and Suggestions for Authors
The paper deals with an interesting issue. It is well written and mathematically sound. However there are some elements that could be improved.
1. How does you approach compare to other works? I understand that there is not much research on arabic but on other languages? The models you use are rather well known do you could compare your results with other sentiment analysis. Such a comparidon cojld provide a really interesting insight into how difficult is sentiment analysis on different languages.
2. I do not know arabic, but you have wriiten that colloquially repeated vowels are used to express emotions. I think that it can be compared to writing in English biiiiiiig instead of big. So do you not miss on some sentiments by neutralizing it?
3. Lines 135-152 look like copy pastel, judt describe the research and tgen simmsrize them together.
Comments on the Quality of English Language
English is ok, paper is rsther easy to read.
Author Response
We thank the reviewer for his pertinent comments. We thank him also for acknowledging the quality of our contributions.
A detailed letter is provided for reviewers addressing each comment. Detailed answers and rebuttals to these comments are provided, and the manner how the article was updated to accommodate them is shown. In the revised version, the modifications suggested by reviewers are highlighted in blue.

Round 2
Reviewer 2 Report
Comments and Suggestions for Authors
The authors addressed all my concerns.
Comments on the Quality of English Language
A new typo occurs in Line 252: “Table 1 provide …”.
Reviewer 3 Report
Comments and Suggestions for Authors
The authors have striven to comply with all my requests. Therefore, from my viewpoint, the paper can be accepted.
Comments on the Quality of English LanguageThe English of the paper appears correct
Reviewer 4 Report
Comments and Suggestions for Authors
The authors have addressed all my issues and I think the paper is now ready for publication.